# Neural Green's Functions

**Seungwoo Yoo**    **Kyeongmin Yeo**    **Jisung Hwang**    **Minhyuk Sung**

KAIST

`{dreamy1534,aaaaa,4011hjs,mhsung}@kaist.ac.kr`

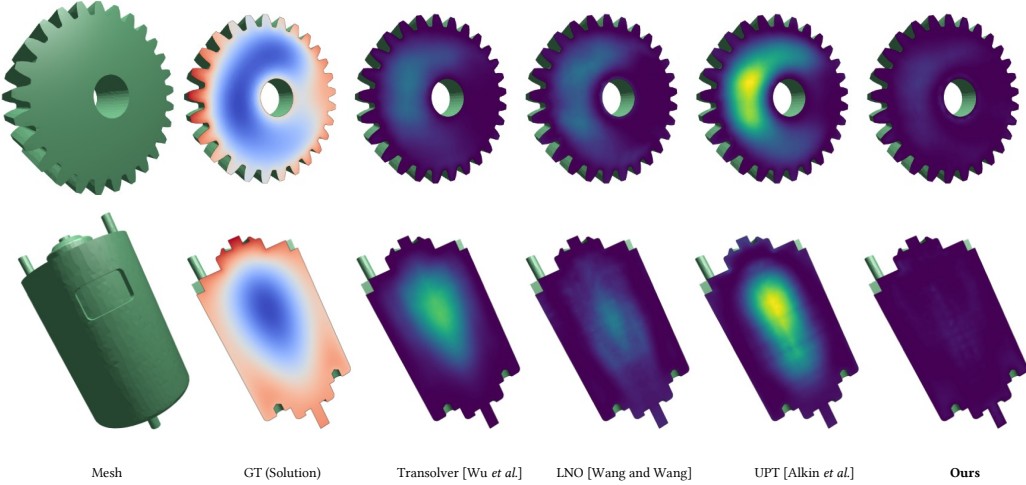

|  |  |  |  |  |  |
|---|---|---|---|---|---|
| Mesh | GT (Solution) | Transolver [Wu *et al.*] | LNO [Wang and Wang] | UPT [Alkin *et al.*] | **Ours** |

**Figure 1: Neural Green's Function.** We propose a neural solution operator designed for linear PDEs whose differential operators admit eigendecompositions. Our framework effectively handles irregular geometries and diverse source and boundary functions. In the steady-state thermal analysis used as a model problem, our method demonstrates superior generalization performance compared to state-of-the-art neural operators in predicting ground-truth solutions (column 2), as evidenced by the error maps (columns 3–6).

## Abstract

We introduce Neural Green's Function, a neural solution operator for linear partial differential equations (PDEs) whose differential operators admit eigendecompositions. Inspired by Green's functions, the solution operators of linear PDEs that depend exclusively on the domain geometry, we design Neural Green's Function to imitate their behavior, achieving superior generalization across diverse irregular geometries and source and boundary functions. Specifically, Neural Green's Function extracts per-point features from a volumetric point cloud representing the problem domain and uses them to predict a decomposition of the solution operator, which is subsequently applied to evaluate solutions via numerical integration. Unlike recent learning-based solution operators, which often struggle to generalize to unseen source or boundary functions, our framework is, by design, agnostic to the specific functions used during training, enabling robust and efficient generalization. In the steady-state thermal analysis of mechanical part geometries from the MCB dataset, Neural Green's Function outperforms state-of-the-art neural operators, achieving an average error reduction of 13.9% across five shape categories, while being up to 350 times faster than a numerical solver that requires computationally expensive meshing.

39th Conference on Neural Information Processing Systems (NeurIPS 2025).

# 1   Introduction

Linear partial differential equations (PDEs) play a central role in many scientific and engineering disciplines, such as thermal analysis, electrostatics, fluid dynamics, and elasticity. Similar to other PDEs, their ubiquity has driven the development of numerical methods, such as finite difference methods (FDMs) and finite element methods (FEMs), to solve the equation defined over complex geometric domains in real-world applications where analytical solutions are often unavailable. These techniques discretize problem domains into fine-grained meshes of lattices or polygons, resulting in linear systems that can be constructed and solved efficiently. However, the reliance on meshes poses a significant limitation, as generating them requires running computationally expensive meshing algorithms [13, 14], making it challenging to rapidly evaluate solutions across multiple problem domains with different boundaries. This bottleneck is especially pronounced during the early design phase of engineering workflows, where rapid iteration is essential for achieving optimal results.

Recently, learning-based solvers have emerged as promising surrogates to conventional numerical solvers for solving PDEs. They predict solutions without mesh construction, either by optimizing a neural network to parameterize the solution function for a given problem instance [30, 19] or by learning solution operators that map input functions to solutions, enabling inference with a single forward pass [20–22, 26, 23, 12, 36, 35, 34, 3]. While these approaches significantly improve efficiency compared to numerical solvers and provide less expensive approximations, little attention has been paid to their generalization capabilities under simultaneous variations in problem domains, source functions, and boundary functions. Some previous methods focus on learning the solutions to PDE problems within a single domain [30, 25, 27], requiring re-training whenever the problem domain or even a specific problem instance changes. Approaches such as Boullé et al. [6], Teng et al. [31], and Negi et al. [28] address this by learning the solution operator for a given domain, enabling the handling of different problem instances. However, they still require re-training when applied to different domains. Others [20–22, 26, 23, 12, 36, 35] attempt to learn solution operators that generalize across a collection of shapes but often struggle to handle unseen source functions and boundary functions. This limitation arises because these inputs are directly provided to the neural networks that regress the solution operators, requiring sufficient training examples to achieve generalization.

In this work, we propose a learnable solution operator capable of handling various shapes and functions, with a specific focus on linear PDEs whose differential operators admit eigendecompositions, such as the Poisson and Biharmonic equations. The foundation of our approach is rooted in the mathematical definition of the solution operator for a linear PDE, known as the *Green's function*, which only depends on the geometry of the problem domain. Once the Green's function of a domain is known, the linearity allows solutions to be computed for arbitrary source and boundary functions. This unique property motivated the design of our framework, which predicts neural features solely from the domain geometry. These features are used to approximate the eigendecomposition of the corresponding Green's function and to predict related differential quantities, such as per-vertex masses, that are essential for computing solutions via numerical integration. This design imposes a strong prior on the solution operator learned by the neural network, making the model independent of the specific source and boundary functions used during training.

In our experiments, we first empirically validate the generalizability of the proposed framework using simple examples of the Poisson and Biharmonic equations. We then extend the evaluation to a practical setting of steady-state thermal analysis on complex 3D geometries, using a benchmark we built from the MCB dataset [16]. This benchmark comprises a diverse collection of mechanical part geometries, each paired with a steady-state thermal distribution obtained by solving Poisson's equation. The dataset features a wide variety of shapes with significant intra-class variations, along with triplets of source, boundary, and solution functions generated by a numerical solver, providing a challenging benchmark setup for both baselines and our method. In comparisons, Neural Green's Function demonstrates superior generalizability compared to state-of-the-art neural operators [35, 34, 3]. Notably, our framework reduces the error metric by 13.9 % compared to Transolver [35], while sharing the same backbone. These results highlight the significance of explicitly formulating and predicting solution operators in achieving superior generalization capabilities.

# 2   Related Work

**Physics-Informed Neural Networks (PINNs).**   Physics-Informed Neural Networks (PINNs) [30] parameterize the solution of each PDE instance directly using a neural network, and optimize its

parameters by minimizing objective functions derived from the governing equations and empirical observations. They can be easily implemented using automatic differentiation, which is supported by deep learning frameworks [29, 2, 7]. However, PINNs are trained separately for each problem instance, with each neural network approximating a single solution at a time. This approach requires retraining whenever the problem domain, source function, or boundary condition changes.

**Neural Operators.** Unlike PINNs, Neural Operators learn function-to-function mappings that act as solution operators for PDEs. They achieve this by parameterizing (nonlinear) kernel functions and implicitly perform kernel integration across subsequent network layers. Specifically, GNO [20] employs graph neural networks (GNNs) to approximate integral kernels through local aggregation via message-passing. FNO [21] improves efficiency and performance by utilizing fast Fourier transform (FFT) [10] in regular domains and learning global integral kernels in the spectral domain. Follow-up work [22, 26] combine GNO [20] and FNO [21] to handle irregular problem domains by efficiently modeling both local and global interactions within systems. Several work [24, 23, 12, 36] utilizing Transformers [33] architecture have also been proposed, leveraging self-attention mechanism to capture interactions among mesh points. These work bypass the quadratic complexity of self-attention layers using linear Transformers [18, 9, 8]. Recent work further reduce the computational complexity by incorporating compact latent representations with dimensionalities significantly smaller than mesh sizes [35, 34, 3]. Neural operators are a versatile framework for operator learning over irregular geometries with diverse source and boundary functions. However, they often couple input meshes with sampled function values, which makes generalization to new functions challenging.

**Learning Green's Functions.** Several recent work focus on linear PDEs and propose data-driven approaches to discovering the Green's function of a problem domain, enabling solutions for varying source and boundary functions. Boullé et al. [6] propose a method to learn Green's functions for linear PDEs using pairs of forcing and solution functions, leveraging rational neural networks [5]. Teng et al. [31] extend this approach by enabling the evaluation of solutions with arbitrary source and boundary functions, regressing Green's functions through the approximation of Dirac delta functions with Gaussian distributions. Similarly, Negi et al. [28] approximate free-space Green's functions using a radial basis function (RBF) kernel-based neural network. By directly learning Green's functions, the aforementioned approaches can predict solutions for PDEs with varying source and boundary functions. However, these methods are restricted to individual problem domains and require retraining for unseen geometries, as the networks learn domain-specific Green's functions and their gradients. Additionally, they focus on simple domains (e.g., circular domains in 2D), where numerical quadrature for computing solutions with predicted Green's functions is relatively straightforward. Our work addresses these limitations through a framework design inspired by the eigendecomposition of Green's functions for linear PDEs. In particular, the framework learns to extract geometric features from problem domains and composes them to reconstruct the corresponding Green's functions. Moreover, our approach predicts differential quantities needed to approximate numerical integration over complex, irregular geometries, extending its applicability to real-world scenarios, as demonstrated in Sec. 5.

## 3 Background

Consider a linear PDE subject to Dirichlet boundary conditions defined over a continuous domain $D \subset \mathbb{R}^d$ with boundary $\partial D$:

$$\begin{cases} \mathcal{L}u(x) = f(x) & x \in D \\ u(x) = h(x) & x \in \partial D, \end{cases} \tag{1}$$

where $\mathcal{L}$ is a linear differential operator, and $u$, $f$, and $h$ are functions that reside in Banach spaces $\mathcal{U}$, $\mathcal{F}$, and $\mathcal{H}$ of functions, respectively. The solution operator $\mathcal{G}$ for Eqn. 1 that computes the solution $u = \mathcal{G}(D, f, h)$ is the mapping:

$$\mathcal{G} : \mathcal{D} \times \mathcal{F} \times \mathcal{H} \to \mathcal{U}, \tag{2}$$

where $\mathcal{D}$ is a set of problem domains. Since Eqn. 1 is a linear PDE, $\mathcal{G}$ is explicitly given as:

$$\begin{aligned} u(x) &= \mathcal{G}_D(f, h) \\ &= \int_D G_D(x, y) f(y) \, dy \\ &+ \int_{\partial D} h(y) \nabla_y G_D(x, y) \cdot n(y) dy, \end{aligned} \tag{3}$$

where $\mathcal{G}_D \coloneqq \mathcal{G}(D, \cdot, \cdot)$ is an instantiation of $\mathcal{G}$ in the domain $D$ and $n(y)$ is the outward normal vector at $y \in \partial D$, respectively. The integral kernel $G_D : D \times D \to \mathbb{R}$, referred to as the *Green's function* of the operator $\mathcal{L}$ in $D$, represents its impulse response and is a solution of:

$$\begin{cases} \mathcal{L}G_D(x,y) = \delta(x-y), & x \in D \\ G_D(x,y) = 0, & x \in \partial D \end{cases} \tag{4}$$

where $\delta$ is the Dirac delta function. One important property of $\mathcal{G}_D$ and Green's function $G_D$ is that they only depend on the geometry of the domain $D$, regardless of the specific source function $f$ or boundary function $h$. This implies that, once the Green's function $G_D$ for a given domain $D$ is known, the PDE in Eqn. 1 can be solved for arbitrary source and boundary functions $f$ and $h$. However, solutions based on Eqn. 3 have only been applied to simple domains where closed-form expressions for Green's functions are available.

Numerical solvers, on the other hand, utilize a volumetric mesh representing $D$ to discretize the operator $\mathcal{L}$ in Eqn. 1, forming a discrete counterpart represented as linear systems. Assume that $D$ is represented as a mesh $\mathbf{D} = (\mathbf{V}, \mathbf{T})$ of a single connected component, consisting of vertices $\mathbf{V}$ and faces $\mathbf{T}$. We define a selection matrix $\mathbf{S} \in \{0,1\}^{N_b \times N_v}$ to indicate $N_b$ boundary vertices among the $N_v$ vertices, and its complement $\mathbf{K} \in \{0,1\}^{(N_v - N_b) \times N_v}$ to represent the remaining interior vertices. The scalar-valued source function $f$ and the boundary function $h$ are discretized by sampling their values at the vertices, resulting in two vectors: $\mathbf{f} \in \mathbb{R}^{N_v}$ and $\mathbf{h} \in \mathbb{R}^{N_b}$, respectively. Then, Eqn. 1 can be written as a linear system in its discrete form:

$$\mathbf{Lu} = \mathbf{Mf} \qquad \text{s.t} \quad \mathbf{Su} = \mathbf{h}, \tag{5}$$

where $\mathbf{L} \in \mathbb{R}^{N_v \times N_v}$ is the discretization of $\mathcal{L}$ and $\mathbf{M} \in \mathbb{R}^{N_v \times N_v}$ is the mass matrix of the mesh $\mathbf{D}$, respectively. The solution to this system can be written as:

$$\begin{aligned} \mathbf{u} &= \mathbf{K}^T \left\{ \left( \mathbf{KLK}^T \right)^{-1} \left( \mathbf{KMf} - \mathbf{KLS}^T \mathbf{h} \right) \right\} + \mathbf{S}^T \mathbf{h} \\ &= \mathbf{K}^T \left\{ \mathbf{G} \left( \mathbf{KMf} - \mathbf{KLS}^T \mathbf{h} \right) \right\} + \mathbf{S}^T \mathbf{h}, \end{aligned} \tag{6}$$

where $\mathbf{G} = \left( \mathbf{KLK}^T \right)^{-1} \in \mathbb{R}^{(N_v - N_b) \times (N_v - N_b)}$ represents the inverse of the matrix $\mathbf{L}$ restricted to the interior of the domain. The detailed derivation can be found in the Appendix. The matrix $\mathbf{G}$ serves the same role as the Green's function discussed in Sec. 3. Notably, the terms $\mathbf{G}$, $\mathbf{M}$, and $\mathbf{K}$ in Eqn. 6 are independent of the given functions $\mathbf{f}$ and $\mathbf{h}$, analogous to the Green's function $G_D$ in Eqn. 3. This independence implies that the solution operator for Eqn. 5 can be approximated by predicting these terms using only the geometric information of the mesh $\mathbf{D}$.

Assuming that $\mathbf{L}$ is symmetric, its submatrix $\mathbf{KLK}^T$, obtained by eliminating the rows and columns corresponding to the boundary vertices, is also symmetric. Furthermore, if the imposed boundary conditions are sufficient, $\mathbf{KLK}^T$ becomes full-rank and admits the following eigendecomposition:

$$\mathbf{G} = \mathbf{\Phi} \mathbf{\Lambda}^{-1} \mathbf{\Phi}^T, \tag{7}$$

where $\mathbf{\Phi} \in \mathbb{R}^{(N_v - N_b) \times (N_v - N_b)}$ is the matrix whose columns are the eigenvectors of $\mathbf{KLK}^T$, and $\mathbf{\Lambda} \in \mathbb{R}^{(N_v - N_b) \times (N_v - N_b)}$ is the diagonal matrix of positive eigenvalues corresponding to the eigenvectors in $\mathbf{\Phi}$. These properties form the basis of our novel framework, designed to predict solutions of linear PDEs across diverse irregular geometries, as detailed in the following section.

## 4 Neural Green's Function

Numerical solvers are well-established tools for solving linear PDEs; however, they rely on the volumetric mesh $\mathbf{D}$ generated from surface representations (e.g., boundary representations in CAD), which can be computationally expensive. This poses a challenge to rapid iteration, which is essential in the early stages of design exploration.

Directly applying neural surrogate models [20–22, 26, 24, 23, 12, 36, 35] may help mitigate this issue by enabling coordinate-based solution queries, eliminating the need for the mesh $\mathbf{D}$. These models effectively capture complex behaviors of nonlinear systems by encoding both query points, along with source and boundary functions, into learned latent representations. However, for the class of linear PDEs discussed in Sec. 3, we claim that a stronger prior can be incorporated into the

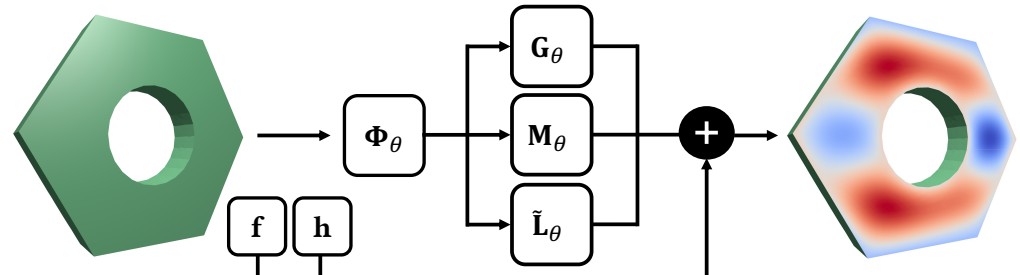

**Figure 2: Method overview.** Given query points representing the geometry of the problem domain where the solution function is to be predicted, Neural Green's Function extracts neural features that are subsequently used to approximate the Green's function and to predict the differential quantities required to evaluate Eqn. 6. The example is rendered using a tetrahedral mesh, which is used solely for visualization purposes.

operator design to improve its generalization capability. In the following, we discuss how the design of Neural Green's Function, inspired by Green's functions, enables generalization to unseen source and boundary functions. Although we mainly focus on problems in 3D space, our approach is not limited to a particular dimension.

Our framework, as outlined in Fig. 2, takes a set of query points $\mathbf{Q}$ in 3D space that represents the geometry of the domain $D$. These points can be obtained through interior point queries [4], thereby avoiding the costly domain meshing. From these query points, we extract features that are used to approximate Green's functions at the corresponding points, rather than directly predicting the solutions. While the query points are associated with a source function $\mathbf{f}$ evaluated at $\mathbf{Q}$ and a boundary function $\mathbf{h}$ sampled at the boundary points, our network remains agnostic to these functions. This independence is the key for achieving generalization to unseen functions.

Specifically, we employ a neural network with parameters $\theta$ to predict features $\mathbf{\Phi}_\theta \in \mathbb{R}^{|\mathbf{Q}| \times d}$ for all query points, whose coordinates are provided as inputs to the network. These predicted features are then utilized to construct our neural Green's function $\mathbf{G}_\theta$, which serves as a drop-in replacement for the ground-truth Green's function $\mathbf{G}$ in Eqn. 6. Inspired by Eqn. 7, we compute each element of $\mathbf{G}_\theta$ as the dot product of feature vectors predicted for two interior points. The computation of $\mathbf{G}_\theta$ for all pairs of points is efficiently parallelized through a single matrix multiplication:

$$\mathbf{G}_\theta = (\mathbf{K}\mathbf{\Phi}_\theta)(\mathbf{K}\mathbf{\Phi}_\theta)^T . \tag{8}$$

Since the mass matrix $\mathbf{M}$ and the submatrix $\mathbf{K}\mathbf{L}\mathbf{S}^T$ of the operator $\mathbf{L}$ are unavailable without a mesh $\mathbf{D}$, we also predict these quantities to evaluate Eqn. 6. In particular, we employ an MLP to decode per-vertex features $\mathbf{\Phi}_\theta$ into per-vertex mass values $\mathbf{M}_\theta$, using the Softplus activation to ensure that the predicted masses are positive. Similarly to $\mathbf{G}_\theta$, the submatrix $\tilde{\mathbf{L}} = \mathbf{K}\mathbf{L}\mathbf{S}^T$ is predicted by computing dot products between features $\mathbf{\Psi}_\theta \in \mathbb{R}^{|\mathbf{Q}| \times d}$, which are decoded from $\mathbf{\Phi}_\theta$ by another MLP:

$$\tilde{\mathbf{L}}_\theta = (\mathbf{K}\mathbf{\Psi}_\theta)(\mathbf{S}\mathbf{\Psi}_\theta)^T . \tag{9}$$

Lastly, these components are combined to evaluate Eqn. 6:

$$\mathbf{u}_\theta = \mathbf{K}^T \left\{ \mathbf{G}_\theta \left( \mathbf{K}\mathbf{M}_\theta \mathbf{f} - \tilde{\mathbf{L}}_\theta \mathbf{h} \right) \right\} + \mathbf{S}^T \mathbf{h}, \tag{10}$$

yielding the predicted solution $\mathbf{u}_\theta$.

Given a dataset of $N$ examples $\left\{ \left( \mathbf{u}^i, \mathbf{f}^i, \mathbf{h}^i \right) \right\}_{i=1}^N$ generated by solving Eqn. 5 using an FEM solver on meshes $\left\{ \mathbf{D}^i \right\}_{i=1}^N$ along with their corresponding mass matrices $\left\{ \mathbf{M}^i \right\}_{i=1}^N$, the network parameters $\theta$ are optimized end-to-end by minimizing the empirical risk:

$$\hat{\theta} = \arg\min_\theta \mathbb{E}_i \left[ \|\mathbf{u}^i - \mathbf{u}_\theta^i\|_2^2 + \lambda \|\mathbf{M}_\theta^i - \mathrm{diag}\left(\mathbf{M}^i\right)\|_2^2 \right], \tag{11}$$

where $\lambda \geq 0$ is a regularization weight applied to the predicted mass values to ensure consistency with the ground truth. We observe that regularizing the predicted masses is essential for the convergence of network training, as discussed further in Sec. 5.4. In all our experiments, we set $\lambda = 1$.

# 5 Experiment

## 5.1 Experiment Setup

In this section, we outline the baselines, evaluation metric, and implementation details of our experiments, while deferring detailed descriptions of the problem setups to their respective sections.

**Baselines.** For 2D Poisson and Biharmonic examples in Sec. 5.2, we compare Neural Green's Function against Transolver [35], the state-of-the-art neural operator. In Sec. 5.3, we further expand the set of baselines to conduct a more comprehensive evaluation in steady-state thermal analysis. This includes Transolver [35], Latent Neural Operator (LNO) [34], and Universal Physics Transformer (UPT) [3]. For all baselines, we use their official implementations and train the networks under the same configuration as ours. For neural operators that require conditioning on source and boundary functions, we follow the approach of Transolver [35] and LNO [34], concatenating the source and boundary function values at the query points as inputs to the baseline models.

**Evaluation Metrics.** Following the baselines [35, 34, 3], the error $e(\mathbf{u}, \mathbf{u}_\theta)$ between the predicted solution $\mathbf{u}_\theta$ and the corresponding ground truth $\mathbf{u}$ is measured as the relative $L_2$ distance:

$$e(\mathbf{u}, \mathbf{u}_\theta) = \frac{\|\mathbf{u}_\theta - \mathbf{u}\|_2}{\|\mathbf{u}\|_2}, \tag{12}$$

which is averaged over the entire test set for quantitative comparisons.

**Implementation Details.** We adopt the network architecture of Transolver [35], a state-of-the-art neural operator and one of our baseline methods, to extract the features $\mathbf{\Phi}_\theta$, encouraging feature sharing across query points. Specifically, our network consists of an MLP that maps coordinates of query points to latent representations. The layer is followed by eight Transolver blocks [35] and MLP heads that decode them into the components required to evaluate Eqn. 10. Unless otherwise specified, all models in our experiments are trained for 40 epochs using the ADAM optimizer [17] with a OneCycleLR learning rate scheduler, setting the maximum learning rate to $1 \times 10^{-4}$. For the steady-state thermal analysis experiments in Sec. 5.3, we use a batch size of 1 and accumulate gradients over 8 training steps to stabilize training, to handle meshes with a large number of vertices.

## 5.2 Two-Dimensional Poisson and Biharmonic Equations

We first assess the generalization capability of Neural Green's Function to unseen source and boundary functions, by fixing the problem domain to a unit square $[0, 1] \times [0, 1]$ in 2D. The domain is discretized with the resolution of $100 \times 100$ by uniformly placing grid points along the two axes. As model problems, we consider Poisson and Biharmonic equations defined over this domain.

Problem instances of Poisson's equation are generated using boundary functions instantiated from the template that satisfies $\Delta u = 0$:

$$u(x, y) = A(x^3 - 3xy^2) + B(y^3 - 3x^2y) + x^2, \tag{13}$$

with the source function set to zero within the domain interior. We use 100 training and 100 test examples, each generated by sampling tuples of coefficients $(A, B)$. To ensure that the boundary functions at test time do not overlap with those used for training, we sample coefficients from the uniform distribution $\mathcal{U}[-1, 1]$ for the training set, and from $\mathcal{U}[1, 2]$ for the test set.

For Biharmonic equation, we consider the template

$$u(x, y) = A(x^4 - 6x^2y^2 + y^4) + Bx^4 + Cy^4, \tag{14}$$

which satisfies $\Delta^2 u = 0$. Analogous to the Poisson's equation setup, we use 100 training and 100 test examples, generated by sampling coefficients over different ranges. For the training set, the coefficients $(A, B, C)$ are drawn from $\mathcal{U}[-1, 1]$, while for the test set, they are sampled from $\mathcal{U}[1, 2]$.

For both setups, we train our Neural Green's Function and Transolver [35] for 40 and 200 epochs, respectively. We increase the number of training epochs for Transolver to account for its slower convergence in training loss. After training, we evaluate both models on the test set by computing the relative $L_2$ error across all examples. As summarized in Tab. 1, Neural Green's Function, which

|  | Poisson's Equation | | Biharmonic Equation | |
|---|---|---|---|---|
|  | Train Set | Test Set | Train Set | Test Set |
| Transolver (Wu et al. [35]) | 0.053 | 0.372 | 0.025 | 0.337 |
| Ours | **0.014** | **0.012** | **0.010** | **0.009** |

**Table 1: Relative $L_2$ errors measured on the training and test sets of 2D Poisson and Biharmonic equation examples.** In both setups, our method achieves comparable errors on the training and test sets, whereas Transolver [35] exhibits a notable gap due to its network being jointly conditioned on both domain geometry and boundary functions.

|  | SCREWS & BOLTS | NUT | MOTOR | FITTING | GEAR |
|---|---|---|---|---|---|
| Transolver (Wu et al. [35]) | 0.221 | 0.320 | 0.407 | 0.180 | 0.281 |
| LNO (Wang & Wang [34]) | 0.239 | 0.372 | 0.528 | 0.259 | 0.466 |
| UPT (Alkin et al. [3]) | 0.358 | 0.516 | 0.765 | 0.392 | 0.507 |
| Ours | **0.189** | **0.275** | **0.338** | **0.160** | **0.243** |
| Error Reduction (%) | 14.7 | 14.1 | 16.9 | 10.8 | 13.3 |

**Table 2: Relative $L_2$ errors measured on the test sets of the steady-state thermal analysis benchmark.** Our method achieves the lowest relative $L_2$ error across all five shape categories. Notably, it improves the metric of Transolver [35] by an average of 13.9%, despite utilizing the same network architecture. The error reduction is the difference between the two errors, divided by the second-best error.

relies solely on features extracted from the domain geometry while remaining agnostic to the specific boundary functions used during training, generalizes well compared to Transolver [35], whose network is jointly conditioned on both domain geometry and boundary functions.

### 5.3 Steady-State Thermal Analysis

Building on the observations from Sec. 5.2, we extend our experiments to a more challenging and practical task, steady-state thermal analysis on complex 3D geometries. This task is appropriate for evaluating the generalizability of our framework to unseen problem domains, beyond the source and boundary functions considered in the previous section. To this end, we construct a new PDE benchmark using the MCB dataset [16], which contains a variety of 3D mechanical part shapes. We generate tetrahedral meshes for shapes in five categories (SCREWS & BOLTS, NUT, MOTOR, FITTING, and GEAR) by meshing the interiors of unit-cube-normalized shapes using fTetWild [14]. The shape collection is divided into 200 shapes for training and 20 shapes for testing to evaluate generalization to unseen problem domains. We illustrate several example shapes in Fig. 3. As shown, the shape collection used in the experiments includes shapes with thin structures and intricate details. Additionally, shapes within the same category can have significantly different geometries, presenting a challenge for learned solution operators to generalize effectively across diverse geometries.

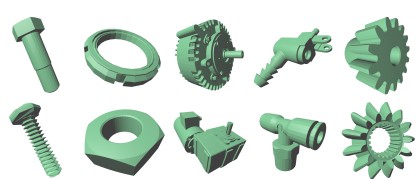

**Figure 3: Example shapes from our dataset.** Our dataset comprises diverse mechanical part shapes from the MCB dataset [16], designed to evaluate the generalizability of learning-based PDE solvers across shape variations.

To generate PDE examples, we employ an off-the-shelf FEM solver [11] to solve Poisson's equation defined on the shapes in our dataset. These problems are constructed using different source and boundary functions derived from predefined templates, details of which are provided in the Appendix. Specifically, we use 8 source functions for training and reserve an additional 8 for testing. For boundary functions, we use 2 for training and 2 for testing. This setup results in 16 unique combinations of source and boundary functions for training, while 16 entirely unseen combinations are used for testing. It offers greater problem diversity than other benchmarks for PDEs on irregular geometries, such as ShapeNet-CFD [32], which assumes a fixed driving velocity. Overall, the dataset comprises 3200 shape-problem pairs for training and 320 pairs for testing across all categories.

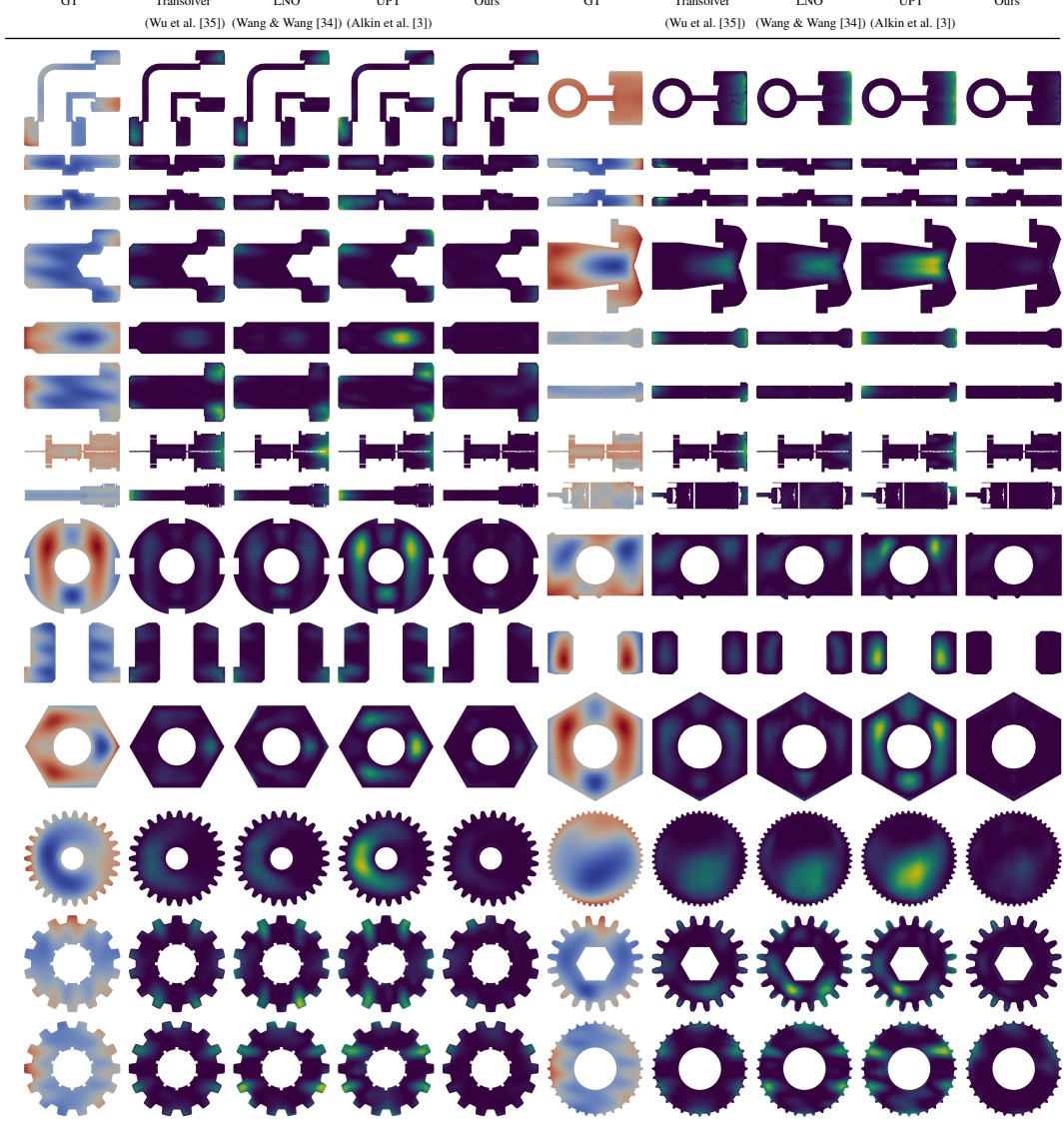

| GT | Transolver (Wu et al. [35]) | LNO (Wang & Wang [34]) | UPT (Alkin et al. [3]) | Ours | GT | Transolver (Wu et al. [35]) | LNO (Wang & Wang [34]) | UPT (Alkin et al. [3]) | Ours |

**Figure 4: Qualitative comparison.** For each ground-truth solution in columns one and six, the per-point $L_2$ errors from various methods are visualized in the error maps across the remaining columns. Red denotes higher values in the columns labeled "GT", while in the error maps, brighter colors indicate greater errors. Neural Green's Function demonstrates its ability to accurately predict solutions on irregular geometries from diverse domains. Compared to state-of-the-art neural operators trained to directly map source and boundary functions to solutions, our method achieves lower errors, as indicated by the darker regions in the error maps. The examples are rendered using tetrahedral meshes, that are used solely for visualization purposes. Best viewed when zoomed-in.

**Quantitative Analysis.** In Tab. 2, we summarize the errors measured on the test sets of five shape categories using the baselines and our method. Neural Green's Function consistently achieves lower errors compared to the baselines, demonstrating the effectiveness of the proposed method that explicitly predicts the solution operator. Compared to Transolver [35] which achieves the second-best errors across all categories, our method achieves an average error reduction of 13.9%, calculated as the difference between the best and second-best errors, divided by the latter. Since our framework utilizes the network architecture of Transolver [35] as its backbone, this improvement highlights the significance of incorporating a prior on the solution operator.

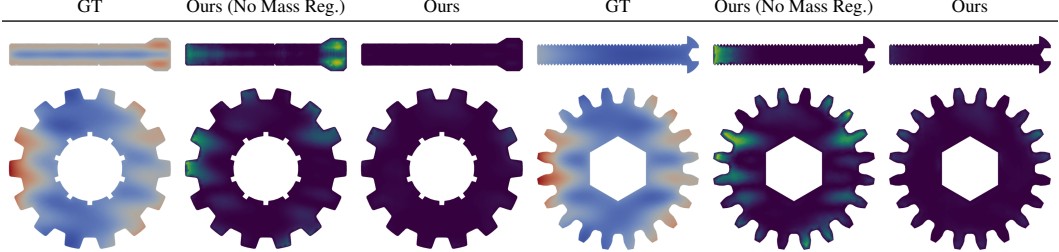

| GT | Ours (No Mass Reg.) | Ours | GT | Ours (No Mass Reg.) | Ours |

**Figure 5: Qualitative results from the ablation study.** For each ground-truth solution in columns one and four, we visualize the per-point $L_2$ errors from variants of our method in the remaining columns. Red denotes higher values in the columns labeled "GT", while in the error maps, brighter colors indicate greater errors. The examples are rendered using tetrahedral meshes, that are used solely for visualization purposes. Best viewed when zoomed-in.

**Qualitative Analysis.** Qualitative results from five categories—FITTING (rows 1–2), SCREWS & BOLTS (rows 3–5), MOTOR (rows 6-7), NUT (rows 8–10), and GEAR (rows 11–13)—are presented in Fig. 4. Additional results can be found in the Appendix. For each example, we display color maps of the ground-truth solutions from the dataset, labeled as "GT", alongside error maps of the predictions generated by different methods. In the solution visualizations, red indicates higher values, while in the error maps, brighter colors represent greater errors. We visualize the slices taken along one of the canonical axes $(x, y, z)$ of each shape for clarity. The darker error maps, compared to those of the baselines, demonstrate that our method produces more accurate predictions, validating our observations from the quantitative analysis. Our method better handles irregular shapes with intricate details, such as thin structures of MOTOR shapes (rows 6-7), or the cogs of GEAR shapes (rows 11-13).

**Runtime Analysis.** We evaluate the efficiency of our framework by comparing it with an FEM solver and neural operator baselines [35, 34, 3], measuring the total runtimes required to process the test sets. The runtimes are summarized in Tab. 6. All reported runtimes are measured in seconds, with each sample processed sequentially to minimize system load and prevent interference that could impact the timing of other samples. Compared to the FEM solver, Neural Green's Function benefits greatly from not requiring mesh generation, which accounts for the majority of the processing time in the FEM pipeline. Notably, it achieves a substantial reduction in runtime, being up to 350 times faster than FEM. Furthermore, Neural Green's Function introduces only minimal computational overhead compared to neural operator baselines [35, 34, 3] while delivering superior generalizability and performance, as discussed previously. Details of this analysis, including the breakdown of the runtimes, are provided in the Appendix.

| | FEM Total (s) | Transolver (s) (Wu et al. [35]) | LNO (s) (Wang & Wang [34]) | UPT (s) (Alkin et al. [3]) | Ours (s) |
|---|---|---|---|---|---|
| SCREWS & BOLTS | 12.956 | 0.033 | 0.022 | 0.056 | 0.039 |
| NUT | 12.238 | 0.022 | 0.020 | 0.076 | 0.051 |
| MOTOR | 50.095 | 0.090 | 0.088 | 0.166 | 0.140 |
| FITTING | 18.439 | 0.032 | 0.030 | 0.076 | 0.059 |
| GEAR | 46.384 | 0.188 | 0.187 | 0.246 | 0.225 |

**Table 3: Runtime comparison against an FEM solver and neural operators.** Our approach achieves up to a $350\times$ speedup over the FEM solver, which requires mesh generation, while maintaining comparable runtime to neural operator baselines and delivering improved generalizability and performance. All runtimes are reported in seconds.

## 5.4 Ablation Study

We analyze the impact of mass regularization during training and the effect of feature dimensionality on model performance. In Tab. 4, we report the test errors of our model and its variant trained without mass regularization, denoted as "Ours (No Mass Reg.)", on the SCREWS & BOLTS and GEAR categories. As reflected in the errors, regularizing the mass prediction outputs from our model

|  | SCREWS & BOLTS | GEAR |
|---|---|---|
| Ours (No Mass Reg.) | 0.285 | 0.411 |
| Ours | **0.189** | **0.243** |

**Table 4: Quantitative results from the ablation study.** Regularization of the mass predictions is crucial for performance.

| $d$ | 64 | 128 | 256 |
|---|---|---|---|
| Relative $L_2$ | 0.180 | 0.189 | 0.206 |

**Table 5: Analysis of feature dimension.** The model's performance remains consistent across different feature dimension choices.

is crucial for performance. This is because the per-vertex masses involved in Eqn. 10 are typically very small (approximately on the order of $1 \times 10^{-4}$), leading to instability during the early stages of network training when the initial mass predictions deviate significantly in scale from the ground-truth values. Qualitative results showcased in Fig. 5 further support this observation: the model trained with mass regularization produces more accurate predictions, as evidenced by the darker regions in the error maps (columns 2–3 and 5–6).

On the other hand, we examine whether the model's performance is sensitive to the dimensionality $d$ of the feature representation $\Phi_\theta$. Intuitively, $\Phi_\theta$ plays a role of the eigenvectors in Eqn. 7, serving as low-rank bases for approximating the operator matrix. To assess this, we train variants of our framework, which by default uses 128-dimensional feature vectors, with alternative configurations employing 64 and 256 dimensions. All models are trained on the SCREWS & BOLTS class using the same setup as in our main experiments, and the relative $L_2$ errors on the test set are reported in Tab. 5. As shown, the model's performance remains insensitive to the choice of feature dimension, highlighting that even 64 learned bases are sufficient to accurately approximate the solution operator.

# 6 Conclusion

We present Neural Green's Function, a solution operator specialized in linear PDEs whose differential operators admit eigendecompositions, capable of generalizing across diverse domains, source and boundary functions. Our framework incorporates the principles of Green's functions into its design by extracting neural features solely from the geometries of problem domains. These features are then used to approximate the corresponding Green's functions and related differential quantities, which are subsequently utilized to perform numerical integration for solution prediction. We empirically validate our design choice by applying Neural Green's Function to solve the Poisson and Biharmonic equations on 2D domains, and further demonstrate that it outperforms state-of-the-art neural operators on a challenging 3D benchmark encompassing a wide variety of shapes and Poisson equation instances defined over them. In particular, Neural Green's Function achieves a 13.9% improvement over the best performing baseline, while accelerating solution evaluation by $350\times$ in steady-state thermal analysis involving complex 3D geometries. These results highlight the superior generalizability of our framework across problem domains, as well as diverse source and boundary functions.

**Limitations and Future Work.** While we believe our approach represents a step toward developing generalizable, data-driven solution operators for PDEs, it is not without limitations. In particular, extending our framework to a broader class of linear PDEs and incorporating additional boundary conditions, such as Neumann and Robin, are promising future directions. Moreover, although our framework achieves runtimes comparable to existing neural operator baselines, further acceleration of the numerical integration step during the forward pass would facilitate its practical application.

# Acknowledgments

This work was supported by the NRF of Korea (RS-2023-00209723); IITP grants (RS-2022-II220594, RS-2023-00227592, RS-2024-00399817, RS-2025-25441313, RS-2025-25443318, RS-2025-02653113); and the Technology Innovation Program (RS-2025-02317326), all funded by the Korean government (MSIT and MOTIE), as well as by the DRB-KAIST SketchTheFuture Research Center.

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

## Appendix

In the following, we present the derivation of Eqn. 6 in Sec. A, details of the problem templates used to define the source and boundary functions for the steady-state thermal analysis experiment (Sec. 5.3) in Sec. B, details of the runtime analysis presented in Sec. 5.3 in Sec. C, and additional qualitative results in Sec. D.

## A   Derivation of the Solution for the Linear System

While the derivation of Eqn. 6 from Eqn. 5 is straightforward, we include the step-by-step derivation for completeness.

Let us decompose the solution $\mathbf{u}$ into interior and boundary components using the section matrices $\mathbf{K}$ and $\mathbf{S}$:

$$\mathbf{u} = \mathbf{K}^T\mathbf{u}_{\text{int}} + \mathbf{S}^T\mathbf{h}, \tag{15}$$

where $\mathbf{u}_{\text{int}} \in \mathbb{R}^{(N_v - N_b)}$ denotes the solution of Eqn. 5 within the domain.

As $\mathbf{u}$ satisfies Eqn. 5, we have:

$$\mathbf{L}\left(\mathbf{K}^T\mathbf{u}_{\text{int}} + \mathbf{S}^T\mathbf{h}\right) = \mathbf{M}\mathbf{f}. \tag{16}$$

Expanding the left-hand side and rearranging terms yields:

$$\mathbf{L}\mathbf{K}^T\mathbf{u}_{\text{int}} = \mathbf{M}\mathbf{f} - \mathbf{L}\mathbf{S}^T\mathbf{h}. \tag{17}$$

By left-multiplying $\mathbf{K}$, we obtain

$$\mathbf{K}\mathbf{L}\mathbf{K}^T\mathbf{u}_{\text{int}} = \mathbf{K}\mathbf{M}\mathbf{f} - \mathbf{K}\mathbf{L}\mathbf{S}^T\mathbf{h}. \tag{18}$$

Assuming $\mathbf{K}\mathbf{L}\mathbf{K}^T$ is full-rank, which is ensured by imposing appropriate boundary conditions, $\mathbf{u}_{\text{int}}$ can be written as:

$$\mathbf{u}_{\text{int}} = \mathbf{G}\left(\mathbf{K}\mathbf{M}\mathbf{f} - \mathbf{K}\mathbf{L}\mathbf{S}^T\mathbf{h}\right), \tag{19}$$

where $\mathbf{G} = \left(\mathbf{K}\mathbf{L}\mathbf{K}^T\right)^{-1}$. Substituting this result into Eqn. 15 gives us Eqn. 6.

## B   Details of Problem Templates

When constructing the benchmark used for the steady-state thermal analysis experiment in Sec. 5.3, the source functions $f$ are populated from the following template:

$$\begin{aligned}
f(x, y, z) = \Delta\{&\sin A\pi x \cdot \cos AC\pi y \\
&+ (1 - \cos A\pi x) \cdot (1 - \sin AB\pi y) \\
&+ \sin^2 AD\pi z\},
\end{aligned} \tag{20}$$

with the coefficients $A = \{1.25, 2.5\}$, $B = \{1.5, 3.5\}$, $C = \{1.5, 3.5\}$, and $D = \{1.5, 3.5\}$, resulting in 16 different combinations. Similarly, the boundary functions are generated using the template:

$$\begin{aligned}
h(x, y, z) = &E(x^3 - 3xy^2) + F(y^3 - 3x^2y) \\
&+ (x^2 - z^2),
\end{aligned} \tag{21}$$

using 4 distinct combinations of the coefficients: $E = \{-1.0, 1.0\}$ and $F = \{0.0, 1.0\}$.

## C   Details of Runtime Analysis

In this section, we provide additional details of runtime analysis in Sec. 5.3. We use the binary compiled from the official implementation of `fTetWild` [14] for generating tetrahedral meshes. The implementation supports multi-core processing, and we used the default parameters with an edge length of 0.02 for mesh generation. To populate the query points used for measuring runtimes, we

| | Meshing (s) | Solve (s) | FEM Total (s) | IPQ (s) | Ours Forward (s) | Ours Total (s) |
|---|---|---|---|---|---|---|
| SCREWS & BOLTS | 12.579 | 0.377 | 12.956 | 0.016 | 0.023 | 0.039 |
| NUT | 11.601 | 0.637 | 12.238 | 0.014 | 0.038 | 0.051 |
| MOTOR | 48.661 | 1.434 | 50.095 | 0.082 | 0.058 | 0.140 |
| FITTING | 18.037 | 0.401 | 18.439 | 0.024 | 0.035 | 0.059 |
| GEAR | 45.803 | 0.581 | 46.384 | 0.180 | 0.044 | 0.225 |

**Table 6: Breakdown of runtime comparison with the FEM solver.** The total runtime of the FEM solver includes both the meshing and linear solve times, whereas our runtime is computed as the sum of the time required for the interior point query (denoted IPQ) and the network forward pass. All runtimes are reported in seconds.

| | | Transsolver (Wu et al. [35]) | | LNO (Wang & Wang [34]) | | UPT (Alkin et al. [3]) | | Ours | |
|---|---|---|---|---|---|---|---|---|---|
| | IPQ (s) | Forward (s) | Total (s) | Forward (s) | Total (s) | Forward (s) | Total (s) | Forward (s) | Total (s) |
| SCREWS & BOLTS | 0.016 | 0.017 | 0.033 | 0.006 | 0.022 | 0.040 | 0.056 | 0.023 | 0.039 |
| NUT | 0.014 | 0.008 | 0.022 | 0.006 | 0.020 | 0.062 | 0.076 | 0.038 | 0.051 |
| MOTOR | 0.082 | 0.008 | 0.090 | 0.006 | 0.088 | 0.084 | 0.166 | 0.058 | 0.140 |
| FITTING | 0.024 | 0.008 | 0.032 | 0.006 | 0.030 | 0.052 | 0.076 | 0.035 | 0.059 |
| GEAR | 0.180 | 0.008 | 0.188 | 0.007 | 0.187 | 0.066 | 0.246 | 0.044 | 0.225 |

**Table 7: Breakdown of runtime comparison with neural operators.** The total runtime of each neural operator is computed as the sum of the time required for the interior point query (denoted IPQ) and the network forward pass. All runtimes are reported in seconds.

uniformly sample the 3D space and determine whether each point lies inside the domain using the Fast Winding Number [4] implemented in `libigl` [15]. To ensure a fair comparison with our method, which leverages GPU acceleration during network inference, we re-implemented the solver algorithm from [11] to support GPU acceleration. In particular, we utilize `Cholespy`[1], a GPU-accelerated Cholesky solver, for matrix prefactorization to solve Eqn. 5. The runtime breakdowns corresponding to the results in Tab. 3 are presented in Tab. 6 and 7, respectively, each of which summarizing the time spent in each stage of the FEM solver—including meshing and solving linear systems—as well as, for the neural operators, the time required for interior point queries (IPQ) and network forward passes. All runtimes are measured on a system equipped with an Intel Xeon Gold 6442Y processor with 24 cores and an NVIDIA RTX 3090 GPU with 24 GB of VRAM.

# D    Additional Qualitative Results

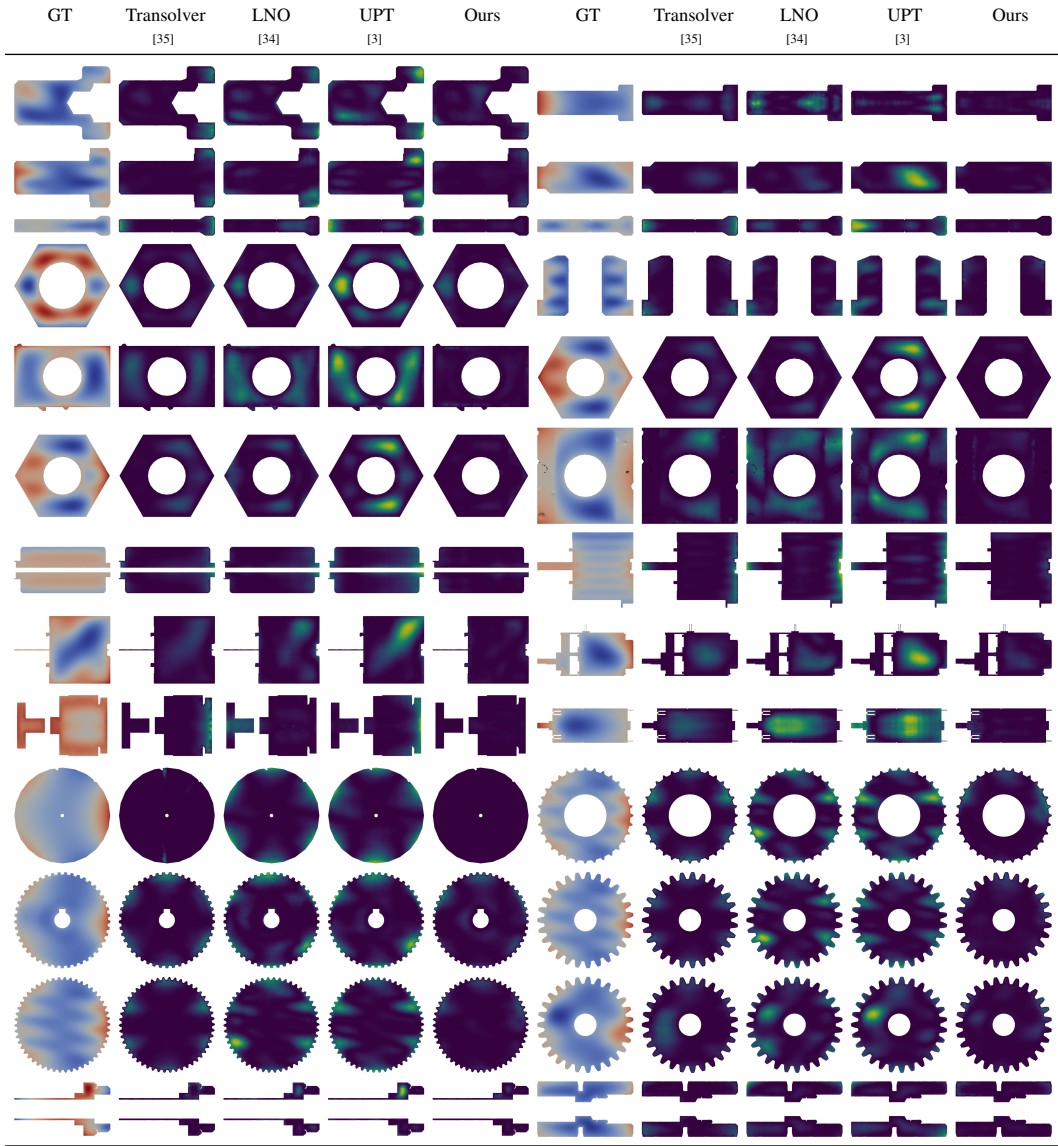

**Figure 6: Additional qualitative comparison.** For each ground-truth solution in columns one and six, the per-point $L_2$ errors from various methods are visualized in the error maps across the remaining columns.

