# OpenReview forum: "Neural Green’s Functions"
_NeurIPS.cc/2025/Conference — NeurIPS 2025 poster_

### Official Review · Reviewer_FSUU · 2025-06-29

**Clarity:** 4
**Significance:** 2
**Originality:** 2
**Rating:** 4
**Confidence:** 5

**Summary:**

This paper offers a neural network based approach to solving Poissons equation with an approach that leverages the computational form of classical linear algebra solution. The main insight is the Shur inverse, the inverse of the interior of the Laplacian, analogous to the discrete Green's function, is approximated with a neural network. This is only possible because of the linearity of the Poisson problem.
Following this the authors make the key insight that the Greens function is only geometry and not boundary value or source term dependent. Given this insight the problem is reduced to predicting Eigen-value scaled Eigen-vectors, a component from the decomposition of the Greens function.

To ensure the method is fully mesh independent this predicted quantity is also used to predict M and KLS.T (the surface to boundary sub-matrix of the Laplacian).

Experiments are performed on 5 complex, non-convex geometries with Neural Green beating Transovler, the next best method, by >10% error reduction, with errors against an FEM approach in the 16-34% range but with a 350 times speed up.

**Questions:**

What kind of neural network is used to predict $\Phi$? In the theme of adding "stronger priors", would it be possible to make a stronger inductive bias for example roto-translational equvariance in this network?

Did the authors try with any exotic feature engineering, ie signed distance function from geometry instead of positional coordinate?

I understand equation (7) inspires the dot products between features, but in (6) G involves the inverse of K.L.K^T. Then how is it justified to use non-verted K/K^T multiplication?

If I could share for idea for the authors to be able to leverage their Poisson specialist method to other PDEs is taking an approach similar to
Zeng et al - "PhyMPGN: Physics-encoded Message Passing Graph Network for spatiotemporal PDE systems", where the Laplace Block calves out the Laplacian solve, deferring other dynamics contributions from parallel NN architectures, GNN (or a NERF if requiring mesh free property).

**Ethical Concerns:**

["NO or VERY MINOR ethics concerns only"]

**Final Justification:**

The authors clarifications, new experiments and including solve times have addressed my concerns.

**Limitations:**

Yes

**Quality:**

3

**Strengths And Weaknesses:**

Formulating the solution operator closely to the linear alegra computational structure is a strong mathematically grounded approach. Allowing to use a stronger inductive bias, utilising the linearity of Poissons equation and the stencil of the numerical method, replacing the computationally complex parts (matrix decomposition) with NN acceleration.

The work is clear, well presented and self contained. The benchmark datasets of the Poisson solves on the MCB dataset is a valuable contribution to the community.

However I have some concerns to the value of the work. A 13.9% improvement is not huge, especially since the errors against an FEM approach still in the 16-34% range. So it's a 1-2% actual error improvement. Moreover the Transolver baseline can be applied to PDEs other than the linear Poisson equation.

Runtime comparison - the paper compares against FEM runtime (which is good) but don't include the runtime of the other ML baselines.
There might be be a sitution where Transolver has a 22.1% error against FEM but a x1000 speedup versus Neural Greens that has a 18.9% error but a x330 speedup.

The authors critise the transformer style baselines saying "but they require large-scale training datasets to achieve generalization."
but I couldn't see any analysis that they overcome the need to for large datasets or any evidence that there method generalizes better than the baselines beyond the 200/20 train/test split.

Although the method is novel, the paper is clear, well explained, my prevailing concerns about the benefits compared to existing neural solvers leave my rating at weak reject unless the authors can convince me on the evaluation and usefulness of this work.

---

> ### Author Rebuttal · Authors · 2025-07-30
>
> Dear Reviewer FSUU,
>
> We sincerely appreciate your detailed review and positive feedback, especially your recognition of our work as a “strong, mathematically grounded approach” presented in a clear and self-contained manner. We have reviewed your questions and summarized our responses below.
>
> **Concerns on Effectiveness & Applicability of Neural Green’s Function**
>
> **Effectiveness.** We would like to highlight that similar magnitudes of absolute improvement (on the order of 1E-3 to 1E-2) were also reported by Transolver, one of the most powerful neural operators to date, on the ShapeNet-CFD dataset, yet were interpreted as meaningful thanks to their substantial relative gains (approximately 20–30%). Following this, we reported relative performance improvements to better reflect the practical significance of our results.
>
> **Application to Other PDEs and Unique Capabilities.** We agree that other neural operators, such as Transolver, are capable of handling a broader range of PDEs. However, we believe that Neural Green’s Function offers distinct advantages due to its design, which closely mirrors the structure of analytical solution operators. In particular, it is independent of the specific test functions used during dataset generation and relies solely on geometry: a property exclusive to ours that we find crucial for generalizing to unseen source and boundary functions. To better support this claim and further demonstrate the generalizability of our framework, we conducted a toy experiment on Poisson’s equation using a fixed geometry with an extended set of boundary function examples.
>
> Specifically, we consider a triangular mesh obtained by discretizing the unit square [0, 1] x [0, 1]. The problem instances were generated using the same template function as in our MCB-B experiments, with the following form used as the boundary condition:
> $$
> f(x, y, z) = A (x^3 - 3xy^2) + B(y^3 -3x^2y) + (x^2 - z^2).
> $$
>
> We generated 100 training and 100 test examples by sampling tuples $(A, B)$. For the training set, all coefficients are sampled from $\mathcal{U}[-1, 1]$, while for the test set, they are sampled from $\mathcal{U}[1,2]$. Each sample, along with the associated triangular mesh, was used to train both Neural Green’s Function and Transolver. During training, we observed that Transolver required a larger number of epochs to reach comparable accuracy on the training set. Accordingly, we trained our model for 40 epochs, while the Transolver was trained for 200 epochs.
>
> After training, we evaluated both models on the test set by measuring the relative $\mathcal{L}_2$ errors, as summarized in the table below. We note that both Transolver and Neural Green’s Function converge on the training set, as evidenced by their low training errors. However, Transolver shows a decline in performance on unseen examples compared to the training set, which may be attributed to the increased difficulty of generalizing to unseen boundary functions. This is due to the joint conditioning over both geometry and boundary function values. In contrast, our method remains agnostic to specific boundary functions and relies solely on geometric features, achieving similar accuracy on the test set.
>
> | Method              | Transolver | Ours   |
> |---------------------|------------|--------|
> | Relative $\mathcal{L}_2$ (Train) | 0.0529     | 0.0139 |
> | Relative $\mathcal{L}_2$ (Test)  | 0.3716     | 0.0124 |
>
> **Application to Other PDEs.** We demonstrate that the proposed framework can be extended to other PDEs, such as the biharmonic equation, as long as their differential operators admit an eigendecomposition. Following the previous experiment setup, we constructed a toy example in which both Transolver and our method were tasked with learning the solution operator for the biharmonic equation on a 2D domain. For the biharmonic cases, we consider the following template function
> $$
> f(x, y, z) = A(x^4 - 6x^2 y^2 +y^4) + B(x^4 - 6x^2z^2 + z^4) + C(y^4 - 6y^2z^2 +z^4),
> $$
> whose bilaplacian ise zero. Following the setup of the previous Poisson experiment, we generated 100 training and 100 test samples by varying the coefficients $A$, $B$, and $C$ of the boundary functions. For the training set, coefficients were sampled from $\mathcal{U}[-1, 1]$, while for the test set, they were sampled from $\mathcal{U}[1, 2]$. Both Transolver and our method were trained on the training split—Transolver for 200 epochs and our model for 40 epochs—and evaluated on the test set after convergence. The relative $\mathcal{L}_2$ errors are summarized in the table below. Our Neural Green’s Function demonstrates consistent generalizability in this new PDE class as well, in contrast to Transolver, whose network is jointly conditioned on domain geometry and function examples.
>
> | Method            | Transolver | Ours   |
> |-------------------|------------|--------|
> | Relative $\mathcal{L}_2$ (Train) | 0.0252     | 0.0102 |
> | Relative $\mathcal{L}_2$ (Test)  | 0.3373     | 0.0093 |
>
> **Runtime Comparison**
>
> We agree that the runtime of neural solvers is an important factor for a thorough analysis. Accordingly, we measured the runtimes of all learning-based baselines on the test set used in our main experiments. As shown in the table below, these methods achieve runtimes on the order of fractions of a second, including both interior point querying and the network forward pass. While Transolver achieves the best runtime performance, please note that UPT and Neural Green's Function are already 280 times faster than a numerical solver, enabling new applications such as real-time solution preview.
>
> | Method    | FEM | Transolver | UPT  | Ours |
> |-----------|------|------------|------|------|
> | Time (s)  | 28.02 | 0.07       | 0.12 | 0.10 |
>
> **Clarifications**
>
> **Network Architecture.** As noted in Section 5.1, our network is based on the Transolver architecture; however, we acknowledge that the provided details were insufficient. Specifically, we use the same network as the Transolver baseline, with only minor modifications—namely, the addition of linear layers to transform the extracted features into the quantities used in Eqn. 10 of the main paper. (e.g., $\Phi_{\theta}$, $\Psi_{\theta}$, and $\mathbf{M}_{\theta}$). We will include these details in the upcoming revision.
>
> **Additional Feature Engineering.** Thank you for your suggestion. While we have not yet experimented with incorporating additional geometric information beyond raw point clouds, we agree that leveraging more sophisticated features (e.g, SDFs) could further improve model performance. We appreciate your input and will explore this direction in future work.
>
> **Use of Non-Inverted K/K^T Multiplications.** In our framework, the solution operator $\mathbf{G}$ is directly approximated by $\mathbf{G}_{\theta}$, which is parameterized as dot products between learned features and can be directly used in Eqn. 10 to compute solutions, without matrix inversion. The multiplication of $\mathbf{K}$ and its transpose in Eqn. 8 serves to index the features corresponding to interior points, ensuring that the resulting matrix has the appropriate shape in $\mathbb{R}^{(N_v - N_b) \times (N_v - N_b)}$.
>
> Lastly, we appreciate your recommendation of an inspiring work (PhyMPGN), which seems highly relevant for advancing our approach.

---

> > ### Comment · Reviewer_FSUU · 2025-08-05
> >
> > Dear authors, thank you for reminder the formulation is only dependent on geometry and not forcing function, and furthermore performing the additional Poisson experiment that showis this enables generalisation to an OOD test set where transolver fails. I think this is a nice model property and the experiments will make a more compelling argument for the paper.
> > Thank you for demonstrating the method at least extends to a limited set of other PDEs whose operator admits an Eigen-decomposition.
> > Thank you for including the times, it's good to know the model is inline with transolver/UPT.
> > Thank you for the clarification on the architecture details, I'm confident this can be sharpened in the revision
> > I update my score acordingly.

---

### Official Review · Reviewer_KHba · 2025-06-29

**Clarity:** 3
**Significance:** 2
**Originality:** 2
**Rating:** 3
**Confidence:** 4

**Summary:**

The authors introduce an operator learning framework - Neural Green's function. The idea is to embed Green's functions in the learning procedure for operators to enable data-efficient learning and deal with the issue of poor generalization in unseen domains, boundary functions and source terms. The authors propose approximating the Green’s function for a given problem domain allowing to compute solutions for arbitrary source and boundary functions by numerically evaluating the integral. They demonstrate superior generalization and speed-ups compared to other state-of-the-art approaches on the MCB dataset for Poisson equation.

**Questions:**

1) Figures: I assume the authors have used the same color-bar for all the error plots in Figure 1. A color-bar for the true solution (column 2) and errors (columns 3-6) would be helpful to the reader. The same applies to Figures 3 and 4 in the main text and Figure 6 in Appendix D.
2) Lines 41-42: I do not think it is quite true that neural PDE solvers generally match the accuracy of classical numerical methods. Optimization difficulties can often make it very hard for the neural PDE solvers to reach the accuracy of numerical solvers. I think this sentence needs to be rephrased. Could you elaborate your view?
3) The authors compare the FEM run-time with the Neural Green's function evaluation/inference time, not the training time. I think it is perfectly fine to do so to stress the advantage during inference. However, it is also important to mention the training times and the fact that the major computational load for neural operators is offloaded to the training phase. Hence, the speed-ups during inference do not come without a price. Moreover, the operator frameworks are trained on data and thus have an advantage that they have more information, compared to solving a PDE for any particular instance with an FEM solver, which does not leverage any data from other simulations.  Could you elaborate your view and why you do not mention the training times?

**Ethical Concerns:**

["NO or VERY MINOR ethics concerns only"]

**Final Justification:**

I appreciate the rebuttal of the authors, and working on a new example. It is very difficult to judge how the updated version (or the code) look like, and the additional statement on the solution being an "initial guess" for subsequent numerical solvers also cannot be evaluated in short time, so I will not reconsider my score.

**Limitations:**

Not mentioned at all, no separate limitation section.

**Paper Formatting Concerns:**

Line 92: typo. It should be "These works ..."

**Quality:**

2

**Strengths And Weaknesses:**

Strengths:
1) Approach: The central idea of incorporating Green’s functions into the framework design is quite interesting. Neural green's function is agnostic to source and boundary functions used for training by design and seems to generalize well across diverse geometries.
2) Empirical validation for the Poisson equation: For the Poisson equation, the baseline comparisons with other state-of-the-art architectures and approaches, different source and boundary terms, and different irregular shapes are strong.
3) Writing: The paper is well-written, and is relatively easy to follow.

Weaknesses:
1) Reproducibility: The code and data are unavailable. The details mentioned in the paper are certainly not enough to reproduce the results. For instance, the architectural details (number of layers, width) are missing, along with many other parameters such as number of interior and boundary points, activation functions, etc.
2) Lack of examples: The study is restricted to a single PDE - Poisson equation. While the Poisson equation is explored in depth, an additional PDE like the Helmholtz equation (for instance, from reference [30] in the paper) would strengthen the results further.
3) Limitations: The limitations of the proposed approach are not mentioned at all. I believe it is quite important to mention the limitations and key directions for future work that the authors feel are important.
4) Weak baselines: the FEM solver is "off the shelf" (l.222), so not necessarily state of the art. Comparing the inference time of a (trained) neural operator in specific examples to an arbitary (i.e., not specifically designed) FEM solver is not reasonable. Poisson's equation is very well known in the scientific community and can be solved extremely efficiently (e.g. using multigrid). An off-the-shelf FEM solver probably cannot compare to a solver designed for this problem. It is ok to use it to generate the training data, but claiming superior performance in terms of inference time is not reasonable (but a very common mistake in physics-informed ML).

---

> ### Author Rebuttal · Authors · 2025-07-30
>
> Dear Reviewer KHba,
>
> Thank you for recognizing the integration of Green’s functions into our framework as a key strength. We also appreciate your positive remarks on the clarity of our writing. Below, we address your comments and summarize our responses accordingly:
>
> **Implementation Details for Reproducibility**
>
> We promise to release the full codebase (including model training and dataset preparation) along with the datasets upon acceptance to support reproducibility and facilitate future research. The network architecture consists of an initial MLP that maps input features (XYZ coordinates and, if applicable, sampled function values) to latent representations. This is followed by eight Transolver blocks and a final MLP that decodes the latent features into either the solution function or, in our case, features used for Green’s function approximation. We use a consistent feature dimension of 128 across all layers.
>
> **Lack of Examples**
>
> We demonstrate that the proposed framework can be extended to other PDEs, such as the biharmonic equation, as long as their differential operators admit an eigendecomposition. To verify this empirically, we prepared a toy setup of the biharmonic equation and compared the performance of our model against Transolver, the strongest baseline in our experiments. The detailed experimental setup and results are described below.
>
> We consider a triangular mesh obtained by discretizing the unit square [0, 1] x [0, 1]. Similarly to our experiment setup involving shapes from the MCB-B dataset, we define a template function of form
> $$
> f(x, y, z) = A(x^4 - 6x^2 y^2 +y^4) + B(x^4 - 6x^2z^2 + z^4) + C(y^4 - 6y^2z^2 +z^4).
> $$
> We use this template function—whose Bilaplacian is zero—to generate our training and test samples by varying the coefficients $A$, $B$, and $C$. For the training set, all coefficients are sampled from $\mathcal{U}[-1, 1]$, while for the test set, they are sampled from $\mathcal{U}[1, 2]$. We generated 100 samples each for training and testing. Along with the corresponding triangular mesh, these samples are used to train both Neural Green’s Function and Transolver. During training, we observed that Transolver required a larger number of epochs to reach comparable accuracy on the training set. Accordingly, we trained our model for 40 epochs, while the Transolver was trained for 200 epochs.
>
> The following table summarizes the training and test accuracies of both models after convergence. While Transolver and our model achieve comparable accuracy on the training set, our approach—being agnostic to the specific boundary and source functions used during training—demonstrates stronger generalization to unseen problem instances. We will incorporate these results in future updates and revise the Methods section accordingly.
>
> | Method              | Transolver | Ours   |
> |---------------------|------------|--------|
> | Relative $\mathcal{L}_2$ (Train) | 0.0252     | 0.0102 |
> | Relative $\mathcal{L}_2$ (Test)  | 0.3373     | 0.0093 |
>
> **Limitations**
>
> While we have demonstrated that our framework extends beyond the originally considered Poisson’s equation, we acknowledge that its applicability could be further improved by supporting a broader range of boundary conditions. We will include a discussion of these limitations in the upcoming revision.
>
> **Clarification on L41-42.** We agree with your point and appreciate your thoughtful feedback, which allows us to clarify and correct our statement. Similar to other neural operator approaches, the primary goal of Neural Green’s Function is to provide an efficient approximation of numerical solvers—potentially useful during the early stages of exploration, before resorting to more precise but more computationally demanding methods. We will revise the relevant text accordingly in the upcoming version.
>
> **Training Time of Learning-Based Solvers** We agree with your point that the acceleration provided by learning-based solvers comes at the cost of training time, making it an important factor to consider when evaluating such methods. In our experiments on the Screws and Bolts and Gear classes from the MCB-B dataset—each containing point clouds with average sizes of 8.6K and 19.8K points, respectively—both our method and Transolver require approximately 15 and 30 minutes, respectively, to process 3,200 examples per training epoch. Since all models are trained for 40 epochs in our experiments, the total training time amounts to approximately 10 hours for the *Screws and Bolts* category and 20 hours for the *Gear* category.
>
> **Clarification on Baseline Setup**
>
> We would like to clarify the FEM baseline setup, particularly regarding the runtime evaluation. As detailed in Appendix C, the baseline was constructed using a state-of-the-art tetrahedralization algorithm [1] and a GPU-accelerated linear solver based on Cholesky decomposition [2], to ensure fair comparison. We will include a more detailed description of the FEM baseline in the main paper in a future revision.
>
> **Color-Bar in Figures.** Thank you for notifying us. As you correctly pointed out, the color bars for the ground-truth solutions (second column) and the error maps (third to sixth columns) in Fig. 1 are shared within each example (i.e., per row). The same convention applies to other visualizations, such as Figs. 3, 4, and 6. We will revise the figures to make this clearer in the upcoming revision.
>
> **Reference**
>
> [1] Fast Tetrahedral Meshing in the Wild (ACM ToG 2020): https://dl.acm.org/doi/10.1145/3386569.3392385
> [2] cholespy: An easily integrable Cholesky solver on CPU and GPU: https://github.com/rgl-epfl/cholespy

---

> > ### Comment · Reviewer_KHba · 2025-08-04
> >
> > I appreciate the rebuttal of the authors, and working on a new example. It is very difficult to judge how the updated version (or the code) look like, and I will not update my score only based on promises in the rebuttal.
> > The statement in the rebuttal about "the primary goal of Neural Green’s Function [being] to provide an efficient approximation of numerical solvers-potentially useful during the early stages of exploration" is valid, but there is no evaluation of this property in the paper (and it was also not written in this way). The authors would have needed to actually keep solving Poissons PDE on the meshes after computing an initial guess to evaluate if a subsequent iterative numerical solver gains anything from the proposed initial condition.

---

> > > ### Author Response · Authors · 2025-08-06
> > >
> > > Dear Reviewer KHba,
> > >
> > > Thank you for reviewing our rebuttal and providing additional feedback. We believe the rebuttal has addressed some of your earlier concerns (e.g., lack of examples, discussion of limitations, and training time details), and we would like to respond to your latest query as follows.
> > >
> > > **Reproducibility**
> > >
> > > We acknowledge your concern regarding reproducibility and would like to reiterate our promise to release the code and dataset to facilitate future research. As the full rebuttal discussion will be made publicly available after the review process, please consider this an expression of our intent to ensure transparency and reproducibility.
> > >
> > > **Validation of Using Neural Green's Function for Numerical Solver Initialization**
> > >
> > > We appreciate your suggestion and have evaluated the effectiveness of our model’s approximate solution as an initializer for numerical solvers, using the 2D toy experiments presented in our rebuttal to Reviewer FSUU. Specifically, we measured the relative $\mathcal{L}_2$ error between the ground-truth solution and the output of a conjugate gradient (CG) solver, implemented as a part of the [SciPy library](https://docs.scipy.org/doc/scipy/reference/generated/scipy.sparse.linalg.cg.html#cg), while varying the maximum number of iterations from 1 to 100.
> > >
> > > The table below summarizes the errors recorded at every 10th iteration of the CG solver, averaged over 100 samples. As shown, the solver achieves low errors even in the early stages of iteration, highlighting the effectiveness of our model's output as an initialization for numerical solvers. In contrast, without initialization the solver takes nearly 100 iterations to reach a comparable error level.
> > >
> > > | Iteration | with_init | without_init |
> > > |-----------|-----------|--------------|
> > > | 1         | 0.009     | 0.943        |
> > > | 10        | 0.003     | 0.800        |
> > > | 20        | 0.002     | 0.661        |
> > > | 30        | 0.002     | 0.532        |
> > > | 40        | 0.004     | 0.416        |
> > > | 50        | 0.008     | 0.313        |
> > > | 60        | 0.011     | 0.224        |
> > > | 70        | 0.014     | 0.150        |
> > > | 80        | 0.016     | 0.090        |
> > > | 90        | 0.017     | 0.053        |
> > > | 100       | 0.017     | 0.038        |
> > >
> > > Thank you for the additional input for enriching our experimental results. We will incorporate this analysis, including a visual plot of the above table, into the revised version of the manuscript.

---

> > > ### Author Response · Authors · 2025-08-09
> > >
> > > Dear Reviewer KHba,
> > >
> > > We appreciate your participation and feedback in the rebuttal and discussion. While we were unable to incorporate the suggested changes into our manuscript during the discussion period due to the current policy, we will ensure these revisions are included in a future version. As the end of the discussion period is approaching, we would like to make a clarification regarding our previous response.
> > >
> > > As noted in L32–36 and L159–162 of the main paper, our primary goal is to address the challenge of evaluating solutions across domains with varying boundaries without meshing—a bottleneck that is particularly pronounced in the early phases of engineering workflows. Accordingly, the central aim of Neural Green’s Function, and the focus of our evaluations, is to rapidly obtain approximate solutions themselves and assess their closeness to those produced by numerical solvers.
> > >
> > > We agree, as suggested in your post-rebuttal comment, that using the outputs of our framework as initializers for numerical solvers is another valuable application of Neural Green’s Function—one in which our method demonstrates strong performance, as shown in our previous response. Nonetheless, we would like to highlight that our primary focus has been on obtaining approximate solutions directly without meshing.

---

### Official Review · Reviewer_fcRy · 2025-07-02

**Clarity:** 3
**Significance:** 3
**Originality:** 3
**Rating:** 5
**Confidence:** 4

**Summary:**

The paper introduces Neural Green's Function (NGF), a mesh-free neural operator for linear Poisson PDEs with Dirichlet boundary conditions. NGF predicts a low-rank approximation of the Green's kernel through spectral feature learning, uses geometry-only conditioning for inference independence, and employs auxiliary MLP heads for solution evaluation. Tested on 3,400 heat-diffusion problems, NGF achieves 14% lower error than Transformer baselines and 350× speedup over FEM.

**Questions:**

1. What value of feature dimension d is used? How sensitive are results to it?

2. Does the model ever predict negative or near-zero eigenvalues causing numerical issues?

3. How are interior query points chosen during training and inference? Uniform sampling? Poisson-disk?

4. Could NGF be combined with reduced-order models (e.g., POD) to further cut costs?

5. Are there cases where NGF underperforms (e.g., extremely thin shells)? A qualitative failure example would be instructive.

6. How difficult would extension be to Neumann/mixed boundary conditions or other linear elliptic operators (bi-Laplacian, anisotropic Laplacian)?

7. For CAD models containing multiple disjoint solids, would spectral factorization remain stable?

8. Since $G_{\theta}$ is built from learned features, how do authors guarantee $KLK^{\top}$ invertibility and positive mass prediction (beyond Soft-plus non-negativity)?

9. Would performance degrade when evaluating on denser grids than training grids?

---

Overall, This paper presents a reasonable operator-learning framework that applies classical Green’s function analysis to solve Poisson’s equations. While the method is clearly explained and the results are promising on the presented benchmarks, the overall contribution is incremental rather than groundbreaking. Some aspects, such as the limitations to specific PDE types, lack of uncertainty quantification, and limited discussion of generalizability, prevent a higher recommendation. With further empirical validation and broader evaluation, the impact and significance of the work could be strengthened. At its current stage, I consider the contribution modest but relevant for the NeurIPS community.

**Ethical Concerns:**

["NO or VERY MINOR ethics concerns only"]

**Final Justification:**

I am quite positive about this paper and believe it would make a valuable contribution to the field upon acceptance.

**Limitations:**

The paper does not explicitly articulate its limitations, but the proposed method is confined to specific types of partial differential equations (PDEs) and lacks uncertainty quantification, offering only limited discussion on generalizability.

**Paper Formatting Concerns:**

I did not observe major formatting issues.

**Quality:**

3

**Strengths And Weaknesses:**

**Strengths:** Unlike prior neural-operator works that learn a black-box mapping, their method explicitly reconstructs the Green’s kernel from geometry alone and exploits linear superposition. Demonstrates significant speed improvements compared to TetWild + Cholesky method. The paper is well organized as well.

**Weaknesses:** Currently limited to Poisson + Dirichlet cases. Framework assumes single connected components. No discussion of stability/positive-definiteness. Interior-point sampling density is fixed (unspecified).

---

> ### Author Rebuttal · Authors · 2025-07-30
>
> Dear Reviewer fcRy,
>
> We appreciate your recognition of our approach—explicitly reconstructing the Green’s kernel from shapes—as a strength compared to existing learning-based solvers that treat the solution operator as a black-box mapping. We believe this direction can contribute to the development of more interpretable learning-based solvers. Below, we have reviewed your questions and compiled our responses accordingly.
>
> **Extension to Other PDE Problems**
>
> We extended our framework to the biharmonic equation, based on the observation that our formulation is applicable to PDEs whose differential operators admit an eigendecomposition. We empirically verified that our method can approximate the corresponding solution operator using a toy example. The detailed experimental setup and results are provided below.
>
> We consider a triangular mesh obtained by discretizing the unit square [0, 1] x [0, 1]. Similarly to our experiment setup involving shapes from the MCB-B dataset, we define a template function of form
> $$
> f(x, y, z) = A(x^4 - 6x^2 y^2 +y^4) + B(x^4 - 6x^2z^2 + z^4) + C(y^4 - 6y^2z^2 +z^4).
> $$
> We use this template function—whose Bilaplacian is zero—to generate our training and test samples by varying the coefficients $A$, $B$, and $C$ of boundary functions. For the training set, all coefficients are sampled from $\mathcal{U}[-1, 1]$, while for the test set, they are sampled from $\mathcal{U}[1, 2]$. We generated 100 samples each for training and testing. Along with the corresponding triangular mesh, these samples are used to train both Neural Green’s Function and Transolver. During training, we observed that Transolver required a larger number of epochs to reach comparable accuracy on the training set. Accordingly, we trained our model for 40 epochs, while the Transolver was trained for 200 epochs.
>
> After training, we evaluated both models on the test set by measuring the relative $\mathcal{L}_2$ errors, as summarized in the table below. We note that both Transolver and Neural Green’s Function converge on the training set, as evidenced by their low training errors. However, Transolver shows a decline in performance on unseen examples compared to the training set, which may be attributed to the increased difficulty of generalizing to unseen boundary functions. This is due to the joint conditioning over both geometry and boundary function values. In contrast, our method remains agnostic to specific boundary functions and relies solely on geometric features, achieving similar accuracy on the test set. We will incorporate these results in future updates and revise the Methods section accordingly. While the presented experiment demonstrates the potential of our framework to generalize to a broader class of PDEs, handling diverse boundary conditions remains an open challenge. We plan to explore extensions of our framework to address this in future work.
>
> | Method            | Transolver | Ours   |
> |-------------------|------------|--------|
> | Relative $\mathcal{L}_2$ (Train) | 0.0252     | 0.0102 |
> | Relative $\mathcal{L}_2$ (Test)  | 0.3373     | 0.0093 |
>
> **Extension to Shapes with Multiple Components and Stability**
>
> We empirically observed that our model remains effective when applied to shapes with multiple disconnected components. This robustness is attributed to the model’s ability to directly approximate the solution operator while capturing point interactions through dot products between feature vectors. Specifically, we analyzed the *Motor* class in the MCB-B dataset and identified 15 shapes with more than two disconnected components. Using our pre-trained model, we predicted solutions for PDE problems generated with the same coefficient configurations as the test set. Across 240 examples (15 shapes × 16 problems each), the model achieved a relative $\mathcal{L}_2$ error of 0.278.
>
> **Discussion of Numerical Stability**
>
> Since our framework does not require explicit prediction of eigenvalues to compute the terms in Eqn. 10, we have not encountered issues related to negative or near-zero eigenvalue predictions. Similarly, since the model directly approximates the inverse operator of $K L K^T$—which is assumed to exist in our problem setup and during dataset construction—no explicit matrix inversion is required during inference. Accordingly, we have not encountered any issues related to invertibility during training or inference.
>
> **Dimensionality of Feature Dimension**
>
> We agree that examining the sensitivity of our model to the feature dimension is important, especially given that, as you correctly noted, our approach can be interpreted as learning a low-rank approximation of the Green’s function. By default, we used 128-dimensional feature vectors. To assess the impact of feature dimensionality, we trained models with 64 and 256 dimensions on the *Screws and Bolts* class, using the same training setup for 40 epochs. As summarized in the table below, the model’s performance remains insensitive to the choice of feature dimension. We will include this analysis in the upcoming revision.
>
> | Feature Dimension | 64   | 128   | 256   |
> |-------------------|------|-------|-------|
> | Relative L2       | 0.18 | 0.189 | 0.206 |
>
> **Sampling Method for Interior Query Points**
>
> For computing interior points, we first perform uniform sampling within the bounding box—centered at the surface mesh’s centroid—with a slight margin of 10%, and use fast winding number computation to identify interior points. Since our approach is data-driven, excessively dense or sparse query point clouds may present challenges. A practical workaround is to match the density of query points to that of the training data, either by selecting a representative subset that captures the overall shape or by sampling additional interior points using uniform sampling combined with winding-number-based rejection.
>
> **Failure Cases of Neural Green’s Function**
>
> We apologize for not including a discussion of failure cases in the paper. As you rightly pointed out, thin regions in the input geometry can pose challenges during inference. While the rebuttal policy prohibits uploading visuals to OpenReview, we will include a detailed discussion of such failure cases, along with qualitative examples, in the revised version.
>
> **Limitations**
>
> We appreciate you for pointing this out. While we have demonstrated that our framework extends beyond the originally considered Poisson’s equation (e.g., Biharmonic equation), we acknowledge that its applicability could be further improved by supporting a broader range of boundary conditions. We will discuss the limitations of our work, including failure cases, in the upcoming revision.

---

> > ### Comment · Reviewer_fcRy · 2025-08-06
> >
> > The authors have effectively addressed my concerns and even provided a new example to support the handling of biharmonic equations. Overall, I am quite positive about this paper.

---

### Official Review · Reviewer_SXcQ · 2025-07-03

**Clarity:** 4
**Significance:** 3
**Originality:** 2
**Rating:** 5
**Confidence:** 4

**Summary:**

This paper presents a method that learns to predict neural approximate Green's functions for Poisson's equation. The approximate Green's function is a low-rank approximation formed from a learned domain-dependent pointwise feature map. The predicted Green's function yields an approximate solution operator that generalizes under changes in domain shape, boundary conditions, and source function.

**Questions:**

- What do you mean by "leveraging the eigendecomposition of Green's functions [by] reconstructing them from features extracted from problem domains" (111-112)? Are the features the eigenvalues/eigenfunctions? Or some approximation thereof?
- Does your method generalize beyond Dirichlet boundary conditions?
- How does your method compare to boundary element / boundary integral methods? Or Monte Carlo/Walk-on-spheres methods, which also support pointwise queries without requiring mesh generation or a global linear solve? When is it advantageous to learn a proxy solution operator vs., for example, using (possibly denoised) Monte Carlo estimates?
- The method represents the geometry of the domain through the sampled query points. Thus the query points are doing double duty as points at which the Green's function and solution are evaluated, and as an input that conditions the predicted solution. How much does the learned solution depend on the details of this sampling? Ideally, the predicted solution at a fixed point $p$ in the domain should be invariant to the choice of other query points $Q$. Does the choice of query points matter at test time? Only during training? Or are the results nearly invariant to these choices, as one would hope?

**Ethical Concerns:**

["NO or VERY MINOR ethics concerns only"]

**Final Justification:**

The authors' rebuttal addressed most of my points. I also see that they did an experiment with a biharmonic equation. I would still like to see evaluation of solution error in $H^1$ rather than just $L^2$.

**Quality:**

3

**Strengths And Weaknesses:**

The results seem pretty impressive. However, the evaluation is on only a limited number of shapes (test set of 20). Since the main point of the paper is generalization to arbitrary shapes, it might be good to stress-test the method on more shapes (e.g., from a completely different collection or category). I am also not sure what you mean in composing combinations of domain, boundary function, and source function. E.g., a given boundary function is defined on the boundary of a particular domain. What does it mean to test the same boundary function across two different domains?

It might be good to evaluate the results quantitatively using the H^1 metric, rather than just L^2 as in equation (12). Given a source function $f$ in $L^2$, the solution to Poisson's equation should lie at least in H^1 in the interior, and it would be good to ensure that not only do the pointwise values of your approximate solution match the ground truth well, but also that their derivatives match well.

To give contex to your runtime comparison in Table 2, it would be fair to note also the total time used for training. (Though I understand the point that *at test time* for an individual domain, the neural proxy is much faster.)

---

> ### Author Rebuttal · Authors · 2025-07-30
>
> Dear Reviewer SXcQ,
>
> Thank you for your positive feedback and for finding our results impressive. We appreciate your constructive comments, which have helped us improve our work. Below, we address your questions in detail:
>
> **Generalization Across Shape Categories and Boundary Conditions**
>
> We agree that the practicality of Neural Green’s Function would be improved if a model trained on one shape category could generalize across different categories. In this work, similar to our baselines [1,2] that learn solution operators over geometries within a specific category (e.g., cars, airfoils), we focused on a relaxed setting where all shapes belong to the same category but exhibit intra-category variations, as shown in Fig. 5 of the main paper. We acknowledge that enabling inter-category generalization would substantially broaden the method’s applicability, and we plan to explore this direction in future work. Similarly, we agree that extending the method to support diverse boundary conditions is an important and valuable direction for future work. We plan to explore this in more depth, and in the upcoming revision, we will clarify the problem statement to more accurately reflect the current scope and limitations of our approach.
>
> **Training Time**
>
> Using the *Screws and Bolts* class from the MCB-B dataset, which contains point clouds with an average of 8.6K points, our model takes approximately 15 minutes to process 3,200 examples per training epoch. On the *Gear* class, with a higher average of 19.8K points, a single epoch takes about 30 minutes. We also evaluated Transolver with a comparable number of parameters and observed similar runtimes across both classes. Since all models are trained for 40 epochs in our experiments, the total training time amounts to approximately 10 hours for the *Screws and Bolts* category and 20 hours for the *Gear* category. We will include a discussion of the runtime in the revised manuscript.
>
> **Clarification of L111-112**
>
> We apologize for the ambiguity in our previous explanation. Our model takes a point cloud as input and is trained to approximate the eigenvalues and eigenfunctions used to construct the operator defined in Eqn. 8 of the main paper. At inference time, our model does not require performing any eigendecomposition of operators defined over the domain. We will clarify this point in the upcoming revision.
>
> **Comparisons Against Boundary Element/Integral Methods and Monte-Carlo Methods**
>
> Thank you for bringing this to our attention and giving us the opportunity to discuss mesh-less PDE solvers that are closely related to our work. We believe each approach has its own strengths, and we would like to note that Neural Green’s Function is designed to complement existing solvers rather than replace them.
>
> Starting with the Boundary Element Method (BEM), to the best of our knowledge, it requires factorizing a dense system upfront. While this enables efficient solves for multiple boundary conditions on the same domain, the factorization must be recomputed when the domain changes. Monte Carlo and Walk-on-Spheres methods, which have recently gained attention in the graphics community, avoid both meshing and matrix factorization by estimating solutions through pointwise random walks. However, these methods typically require a large number of samples to achieve low-variance estimates, resulting in longer runtimes compared to learned proxy operators like ours, which produce solution estimates in a single forward pass.
>
> Although our method requires an initial training phase, it enables fast inference through a single forward pass—a key distinction from the aforementioned methods. In real-world applications, we envision a hybrid workflow where Neural Green’s Function is used for rapid prototyping or early-stage exploration, providing candidate solutions that can be further refined using more precise solvers such as BEM- or WoS-based approaches.
>
> **Impact of Choosing Query Points**
>
> Our analysis conducted on the test set, summarized in Tab. 1, implies that the model can handle point clouds of varying sizes—and consequently, densities—since all shapes are normalized to the unit cube. However, as our approach is data-driven, excessively dense or sparse query point clouds may present challenges. A practical workaround is to match the density of query points to that of the training data, either by selecting a representative subset that captures the overall shape (if input is dense) or by sampling additional interior points using uniform sampling combined with winding-number-based rejection (if input is sparse).
>
> **Table 1. Statistics of Point Cloud Sizes**
>
> | Category          | $V_{\text{avg}}$    | $V_{\text{std}}$    | $V_{\text{min}}$ | $V_{\text{max}}$  |
> |-------------------|----------|----------|--------|--------|
> | *Nut*               | 18086.10 | 7492.09  | 6849   | 34189  |
> | *Screws and Bolts*  | 8666.00  | 5836.00  | 1733   | 21679  |
> | *Gear*              | 19790.30 | 11614.31 | 5051   | 49634  |
> | *Fitting*           | 13962.35 | 3854.70  | 4978   | 20406  |
> | *Motor*             | 27106.25 | 10352.99 | 9942   | 49454  |
>
> **Composing Combinations of Domain, Boundary Function, and Source Function**
>
> By using the same boundary function across different domains, we meant that the coefficients of the template function—evaluated at the boundary points of each domain—are kept identical. We apologize for the confusion and will revise the text to clarify this aspect of our experimental setup.
>
> **Reference**
>
> [1] Transolver: A Fast Transformer Solver for PDEs on General Geometries: https://arxiv.org/abs/2402.02366
> [2] Universal Physics Transformers: A Framework For Efficiently Scaling Neural Operators: https://arxiv.org/abs/2402.12365

---

### Note · Authors · 2025-08-16

Dear Reviewers and Area Chairs (ACs),

We appreciate your efforts in managing and participating in the review process. We are especially grateful to the reviewers for describing our paper as “well-organized” (fcRy), “easy to follow” (KHba), and “well-presented and self-contained” (FSUU). Based on our discussions, we would like to make the following final remarks regarding the concerns raised during the discussion period:

**Promise on Revision [Reviewer KHba]**: As Reviewer KHba pointed out, we acknowledge that the paper could be further improved by including more implementation details (*e.g.*, network architecture) to facilitate reproducibility. We also commit to releasing the training and inference code, along with pretrained models, together with the revised paper.

**Main Goal of Neural Green’s Function [Reviewer KHba]**: We thank Reviewer KHba for the feedback on our writing and would like to reiterate the clarification provided during the discussion period. As stated in our rebuttal—“the primary goal of Neural Green’s Function is to provide an efficient approximation of numerical solvers, potentially useful during the early stages of exploration…”—and as discussed in L32–36 and L159–162 of the main paper, the aim of Neural Green’s Function is to address the challenge of evaluating approximate solutions of linear PDEs over irregular geometries without meshing, a major bottleneck identified in our analysis summarized in Tab. 4 of Appendix C. To support this, we presented both qualitative and quantitative comparisons between neural PDE solvers, focusing in particular on error against ground-truth solutions.

We will incorporate all items discussed during the rebuttal and discussion period into the upcoming revision, and once again appreciate your feedback.

---

### Decision · Program_Chairs · 2025-09-17

**Decision:**

Accept (poster)

**Comment:**

This paper presents the Neural Green's Function (NGF), a neural operator for linear Poisson PDEs with Dirichlet boundary conditions. The authors demonstrate that NGF enables data-efficient learning, improves generalization across unseen domains and boundary/source functions, and achieves significant speed-ups compared to state-of-the-art approaches.

The majority of reviewers recommended acceptance, highlighting the paper's strong mathematical grounding, clear exposition, and compelling empirical results. The authors also effectively showcase NGF's utility beyond the Poisson equation, making a strong case for its value.

One reviewer (KHba) raised several concerns that I believe the authors have adequately addressed in their rebuttal. They have committed to releasing their code to ensure **reproducibility**, added a new experiment on the biharmonic equation to **broaden the examples**, provided sufficient justification for their **choice of baselines**, and presented promising preliminary results for **initializing numerical solvers**.

After weighing the paper's clear strengths against its minor weaknesses, I recommend **acceptance**. This principled approach, backed by strong results, will be a valuable contribution to the community and will likely inspire further work. I strongly encourage the authors to incorporate the new experiments and valuable discussions from their rebuttal into the final version of the paper.